# Ground-Penetrating Radar Prospections to Image the Inner Structure of Coastal Dunes at Sites Characterized by Erosion and Accretion (Northern Tuscany, Italy)

**Adriano Ribolini *** , **Duccio Bertoni** , **Monica Bini** and **Giovanni Sarti**

Department of Earth Sciences, University of Pisa, Via S. Maria 53, 56126 Pisa, Italy; duccio.bertoni@unipi.it (D.B.); monica.bini@unipi.it (M.B.); giovanni.sarti@unipi.it (G.S.)
* Correspondence: adriano.ribolini@unipi.it

**Abstract:** In this study we aimed to gain insights into dune formation and evolution from select coastal tracts of Northern Tuscany by inspecting their internal sedimentary architecture with Ground-Penetrating Radar (GPR) analysis. Erosion, equilibrium and accretion characterize the selected coastal tracts, and this analysis remarks on some GPR features consistently associated with specific coastal evolution states. A standard sequence of data processing made it possible to trace several radar surfaces and reflectors in the GPR profile, eventually interpreted in terms of depositional processes and erosive events. The stable or currently accreting coastal sectors show radar features compatible with a general beach progradation process, punctuated by berm formation in the general context of a positive sedimentary budget. Additionally, the radar facies distribution locally supports a mechanism of dune nucleation on an abandoned berm. Conversely, the GPR profile of the coastal sector today affected by erosion shows how a negative sedimentary budget inhibited coastal progradation and favored destructive events. These events interacted also with the active dunes, as demonstrated by the overlapping of wave run-up and aeolian radar facies. GPR prospections were effective at delineating the recent/ongoing coastal sedimentary budget by identifying radar features linked to construction/destruction phenomena in the backshore, and to dune nucleation/evolution.

**Keywords:** Ground-Penetrating Radar; radar facies; coastal dunes; beach progradation; berm; coastal erosion; Italy

## 1. Introduction

The internal structures of coastal strand-plains and dunes can offer relevant information with respect to the morphodynamic processes that determined their formation and those that guide their current evolution ([1,2] and reference therein). Therefore, the stratigraphic architecture may represent an archive potentially yielding data about wind intensity/direction, sea-storms' intensity/frequency, sediment availability, incoming wave energy and direction, nearshore geomorphology, shoreline configuration and sea-level changes [3–13]. Nevertheless, obtaining this information has proven extremely difficult because field exposures are frequently limited and preserved only for a short time. Moreover, the non-cohesive nature of aeolian sediments limits artificial trenches to only a few meters below the ground's surface, and the low percentage of sediment recovery makes cores scarcely effective in most cases. Even successfully undertaken, these two approaches offer only a punctual or very restricted vision of the subsurface's stratigraphic architecture, and the potential lateral extension of sedimentary bedding architecture is largely based on interpolations between spotted data and/or subjective spatial interpretation.

Among the non-destructive techniques for subsurface exploration, Ground-Penetrating Radar (GPR from this point ahead) is a geophysical method that may provide imaging of the internal structures of coastal dunes, by obtaining vertical sections of the subsurface several hundreds of meters long and some meters deep, at a (sub)decimetric resolution.

Therefore, many studies have adopted the GPR method to get very detailed images of the internal structures of onshore coastal sedimentary bodies ([14–25] among others).

Sandy beaches backed by dunes are coastal landforms widespread along the microtidal reflective beaches in the Mediterranean [26,27]. The Italian peninsula exhibits more than 3000 km of beach, in many cases characterized by a single or a series of dune ridges [28–33]. Most of these coastal areas are undergoing intense erosion, having shown significant coastline retreat over the last few decades. The sandy beaches of Northern Tuscany (Versilia Plain) do not escape this erosive process, which has affected the area at different rates since the beginning of the 20th century [34]. However, there are some coastal tracts that exhibit an equilibrium or even progradation, making Northern Tuscany a candidate for understanding the delicate balance of erosional/depositional processes governing coastal dynamics.

In this work we aimed to reconstruct the stratigraphical beddings and erosional surfaces composing the internal structures of dunes from selected coastal tracts of Northern Tuscany (Versilia Plain) that are characterized by erosion, accretion and equilibrium (Figure 1). Once the primary sedimentary structures of the examined dunes were deciphered from GPR data, we attempted to use them for reconstructing the major morphodynamic events of the coastal sectors, exhibiting different behaviors.

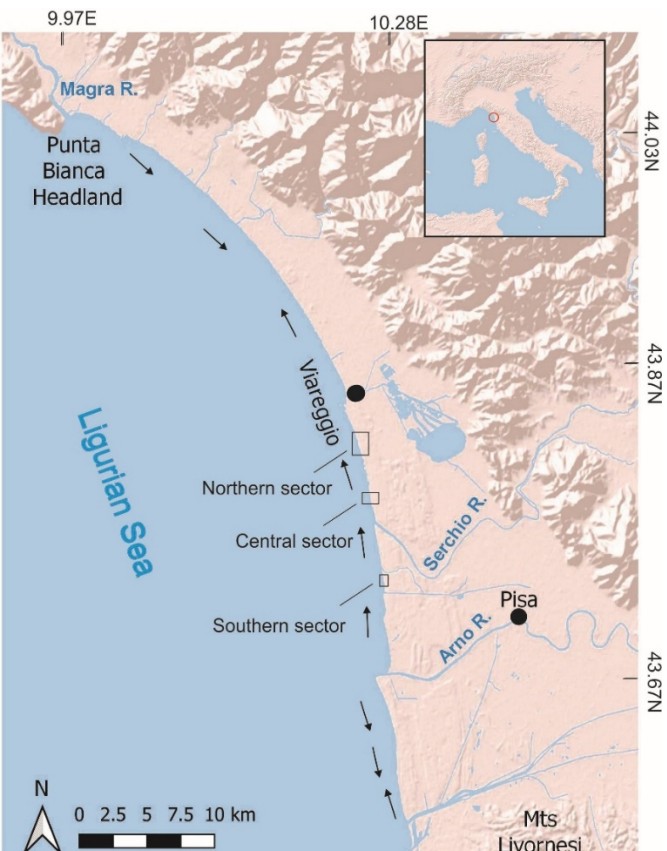

**Figure 1.** Geographical sketch of the Northern Tuscany littoral cell. The black arrows represent the direction of the littoral drift.

## 2. Study Area

### 2.1. Geography, Geomorphology and Sediment Supply

The sector of coast where the surveys were conducted belongs to the Northern Tuscany littoral cell (Figure 1), which is a 65 km long stretch of coast located in the eastern part of the Ligurian Sea (western side of Italy). The cell is within Punta Bianca Headland to the north and bordered by the Mts Livornesi to the south. The backshore is delimited by a dune field, which is only preserved in a few protected areas. This is part of a large

strand-plain that formed during the last 3000 years thanks to the sediments supplied by the main streams of the area [35,36].

The coast was fed by three main sources of sediments, i.e., Arno, Serchio and Magra rivers (Figure 1). While Serchio River sediment supply does not contribute to the natural nourishment of the beach due to the low sediment load (23,000 t/year), Arno (1,524,000 t/year) and Magra (632,000 t/year) rivers feed the southern and northern sectors of the cell respectively [37]. So far, only a very few direct and non-systematic measurements are available [38] and, the present-day natural sediment budget is largely unknown. The littoral drift is directed to the south in the northern sector of the littoral cell, whereas the central sector is characterized by a northward-trending drift that generates at the Arno River mouth [39,40]. The littoral drift is southward-trending south of the Arno River delta. The sea weather is usually characterized by westerly waves, but the strongest storms (wave height higher than 6 m) come from the south-west (Figure 1). The tidal regime is micro-tidal, as the range is hardly over 30 cm [41,42].

### 2.2. Investigated Sites

The fieldwork was carried out at three separate sites (Figures 1 and 2). The northern site is about 3 km south of the Port of Viareggio. The area is characterized by medium sand (0.27 mm) and a large, well-developed backshore, which is about 60 m wide [42–44]. Coastal dunes are constituted by embryonal dunes and frontal dunes about 3 m high and do not show any sign of erosion. Indeed, this sector of the coast is presently accreting due to the constant feeding from the updrift areas, which are eroding [45].

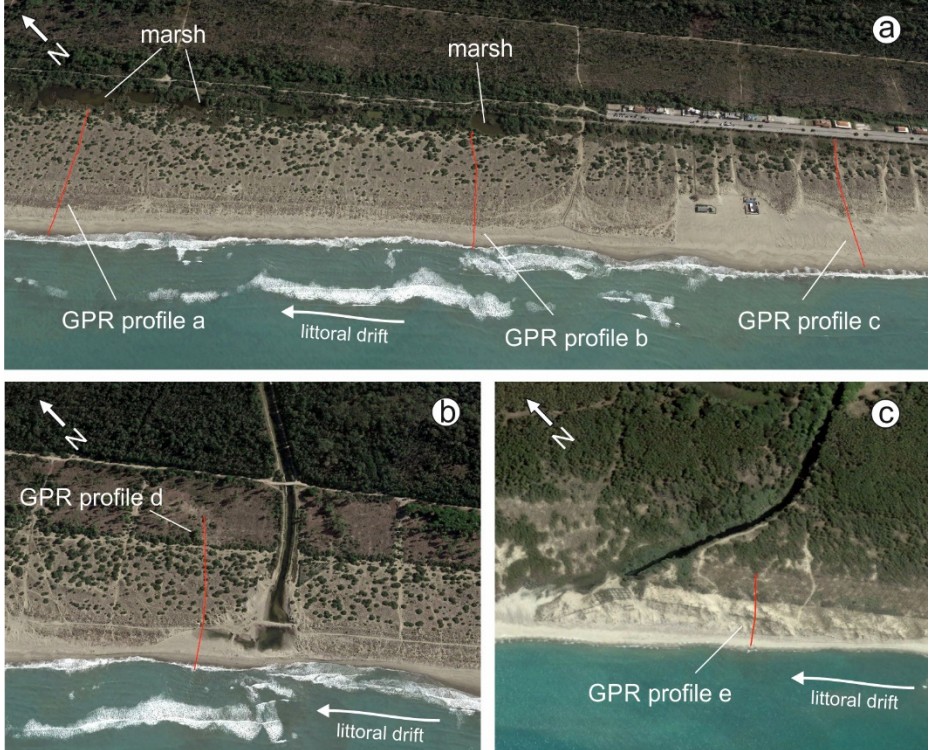

**Figure 2.** Oblique satellite imagines indicating the positions of the GPR profiles in the northern (**a**), central (**b**) and southern (**c**) sectors. For sectors' locations in the littoral cell, see Figure 1. Source, Google Earth; acquisition dates, 2019 (**a**,**b**) and 2014 (**c**).

The central site is about 6 km south of the Port of Viareggio (Figures 1 and 2). The average grain size is about 0.3 mm (medium sand); the backshore is about 60 m wide [46]. The dune system is similar to that of the northern site; frontal dunes are just slightly higher (about 4 m, 43).

The southern site is about 8 km north of the Arno River mouth (Figures 1 and 2). Here, the average grain-size is slightly larger than in the north (about 0.32 mm, medium sand) [46], but the greatest difference is represented by the backshore width, which is about 10 m wide due to strong erosion effects that are quickly obliterating the coastal dunes. Wave processes completely eroded the 6–9 m high frontal dune in recent years [47]. Presently, the area only has semi-mobile dunes.

## 3. Materials and Methods

We used a GPR system provided by IDS Georadar Company© (https://idsgeoradar.com/) equipped with a monostatic transmitter and receiver operating at 600 MHz (nominal peak frequency) via a shielded antenna [48]. To reach greater depths at one site (Figures 1 and 2c, southern sector), we used a monostatic HH-polarized transmitter and receiver operating at 200 MHz (nominal peak frequency) and HH-polarized. In both cases, the transmission and reception antennae were separated by 0.19 m and oriented in broadside mode. We acquired data in continuous mode, controlling the in-line trace spacing (2.4 cm) with an odometer wheel. Each vertical trace was reconstructed by means of 1024 samples taken in a time window of 100 ns. We obtained several GPR profiles orthogonal to the present coastline, each terminating at the beach face, further ahead of the berm ridge/terrace (Figure 2). For each GPR profile, the corresponding GPS trace was recorded with a barometric altimeter (Garmin model, $\pm 3$ m accuracy, 0.3 m resolution). The GPS point corresponding with the coastline was set to 0 m asl and used to calibrate the rest of the profile.

The GPR data processing followed a standard sequence aimed at removing instrumental/environmental noises, gaining weak signals and converting times of recorded reflective events into depths [49]. Finally, we adjusted the GPR profiles to the topographies extracted by the GPS data. We used GPR Slice software (by Geophysical Archaeometry Lab, Los Angeles, CA, USA) to achieve all these processing steps.

For the interpretation of the GPR profile, we decided to follow the approach of the radar stratigraphy, frequently used in coastal/aeolian contexts [8,11,14,18,50,51]. Systematic terminations of reflections were used to identify radar surfaces, which in turn delimited at the top and at the base 2D sets of reflections, referred to as radar facies. The distinctive characteristics of the reflections composing the radar facies (shape, dip, mutual relation and continuity) [14,52] reflect features of the stratigraphical beddings, i.e., stratigraphical facies. Radar surfaces represent non-depositional or erosional hiatuses in a sedimentary sequence. The GPR profiles are reported exaggerating the elevation (vertical) scale compared to the distance (horizontal) scale by a factor of 3.5 or 7 (i.e., $3.5\times$ and $7\times$). The dips of the reflections are reported in the text without any exaggeration ($1\times$) to appreciate their real values.

## 4. Results

### 4.1. Data Processing

After removing undesired frequencies coming from instrumental and environmental noise with a band-pass filter [140-190-500-700], we applied a gain function (Automatic Gain Function—AGC) to enhance the visibility of deeper reflections affected by signal attenuation. We subtracted the mean trace from the dataset to remove continuous flat reflection caused by the ground's surface (background removal). By limiting this last filter to the first 10 ns, the disruption of reflections from continuous flat layers below the surface was avoided. The existence of some diffraction hyperbolas made it possible to use the synthetic hyperbola method [49] to estimate that GPR wave velocities of 9–11 cm ns$^{-1}$ were consistently found across all profiles (Figure 3). We adopted a constant average velocity of 10 cm ns$^{-1}$ for time-to-depth conversion. Finally, we applied a static correction GPR profile using the topographies supplied by the GPS surveys. The low dips of the topographic surfaces made the correction by the antenna tilting useless.

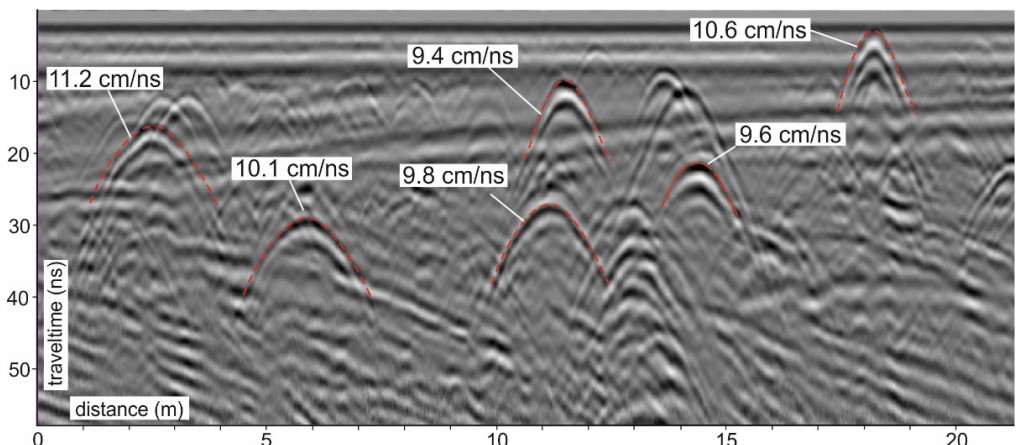

**Figure 3.** Radar profile with some of the diffraction hyperbolas used for the determination of GPR waves' velocities. From the velocity dataset, an average velocity of 10 cm/ns was adopted.

The GPR profiles are presented in color mode (grey tones), with a relative normalization of amplitudes and no linear/exponential graphical gains.

### 4.2. GPR Surface and Facies

According to radar stratigraphy concepts, we detected two types of radar surfaces in the 2D radar profiles (Figure 4) based on reflection terminations, facies differences and the existence or not of a coherent reflection systematically truncating the other reflections. These radar surfaces delimit eight types of radar facies differentiated based on shape, dip, mutual relation and continuity of the reflectors that compose them (Figure 5).

Facies with a seaward inclination predominated, especially in the deeper portions of the GPR profiles. The inclination, length and locally concave/convex shapes allowed us to group the seaward inclined reflectors into two facies, i.e., SI1 (4–6°, 1×) and SI2 (6–8°, 1×). Landward inclined reflectors characterize infrequent and limitedly extensive facies. Two facies were distinguished according to the inclination and reflection amplitude, i.e., LI1 and LI2 (Figure 4).

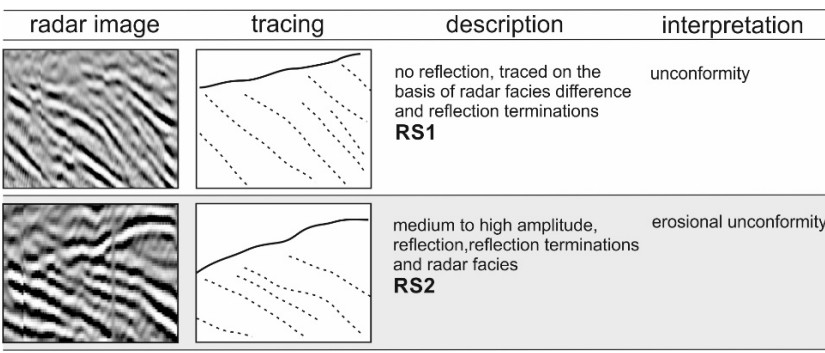

**Figure 4.** Radar surfaces. GPR image, reflections tracing and interpretation.

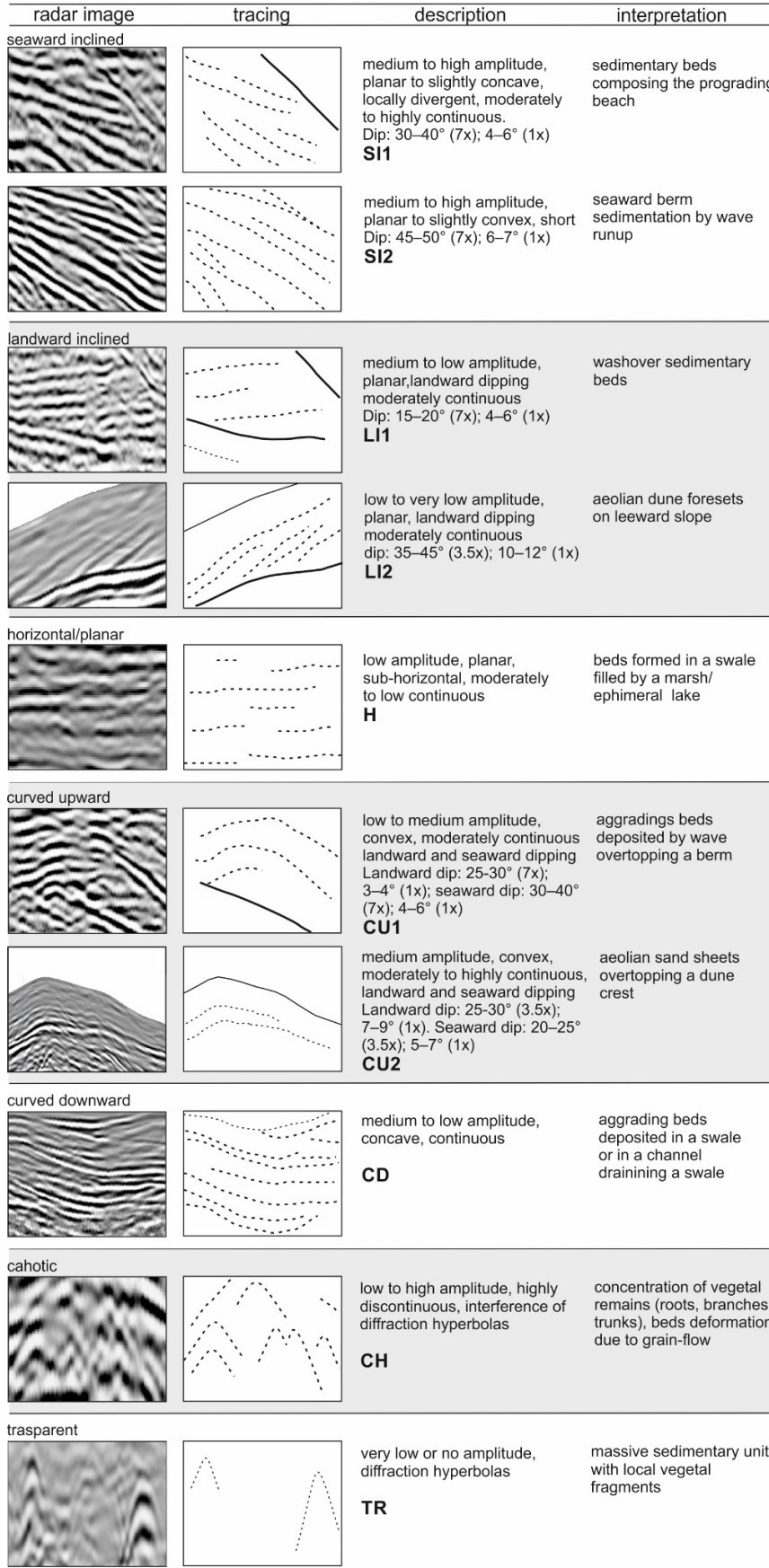

**Figure 5.** GPR image, reflections tracing and interpretation of the radar facies.

Between the two upward curved facies, the one characterized by medium amplitude, short lengths and moderate continuity (CU1) was interbedded between SI1 facies (Figure 5). The downward curved facies (CU2) was composed of more continuous and longer reflectors, with seaward and landward inclinations greater than those of CU1 (5–7° and 7–9°, respectively; 1×). The facies composed of downward curved reflectors (CD) is rarely present in the GPR profiles (Figure 5). Interference phenomena of diffraction hyperbolas explain the presence (even rare) of radar facies with chaotic reflectors, i.e., lacking a coherent geometry (CH).

The reflective framework is completed by a radar facies exhibiting very weak or absent reflections and sporadic phenomena of diffraction (TR) (Figure 5). This facies exclusively characterizes the shallower parts of the subsurface.

### 4.3. GPR Facies Association and Interpretation

Following the radar facies associations and their distribution, the GPR profiles show the existence of two main sedimentary units separated by an erosional unconformity (Figure $6a_1,b_1,c_1,d_1,e_1$). The upper unit is composed of upward curved (CU2) and transparent (TR) radar facies. Curved reflectors are often continuous in the seaward and landward sides, but stratification is well visible only in a dip direction (i.e., perpendicular to the coastline). CU2 facies can be distinguished from the reflectors of other facies (LI1, LI2, SI1) by (i) the lower reflection amplitude, (ii) the wavy shape and (iii) the high angles of the reflectors (up to 30°, see Figure $6a_1,b_1,c_1,d_1,e_1$). These stratigraphic characteristics are coherent with dominantly wind-based sand transport and sedimentation related to coastal dune depositional environment. In particular, the high angle of inclination of the stoss and leeside foresets is typical of modern and ancient aeolian settings [53–55]. In the lower unit, facies with seaward orientation predominate (Figure $6a_1,b_1,c_1,d_1,e_1$). The radar facies are consistent with a prograding strand-plain depositional setting (SI1) characterized by a wide backshore with locally formation and growth of berms (SI2, CU1).

The high angle (6–7°, 1×) mainly seaward-oriented beds reflect the development of storm berms and their migration due to the coastline progradation. The superimposition of low angle (4–6°, 1×) beds with more inclined seaward (6–7°, 1×) and landward orientations records the progressive abandonment of storm berms due to the seaward coastline migration. Landward inclined beds were generated during washover events (LI1) (Figure $6a_1,c_1,d_1,e_1$). The downward curved radar facies (CD) is interpreted as a filler sequence of swale areas. These areas, during the storm phases, can be also worked as draining channels. In the northern and central sectors (Figure 2a,b), the innermost portion of the backshore is characterized by the growth of an incipient dune (*sensu* [56]) colonized by a pioneer vegetation. We interpreted the TR and CH radar facies as being dominated by vegetal remains or sedimentary deformations due to development of roots (Figure $6a_1,b_1,c_1,d_1$).

The formation of a small marshes and ephemeral lakes (see Figure 2 for the present-day evidence), which is common in a strand plain depositional environment, may be represented in the subsurface by low amplitude sub-horizontal facies (H).

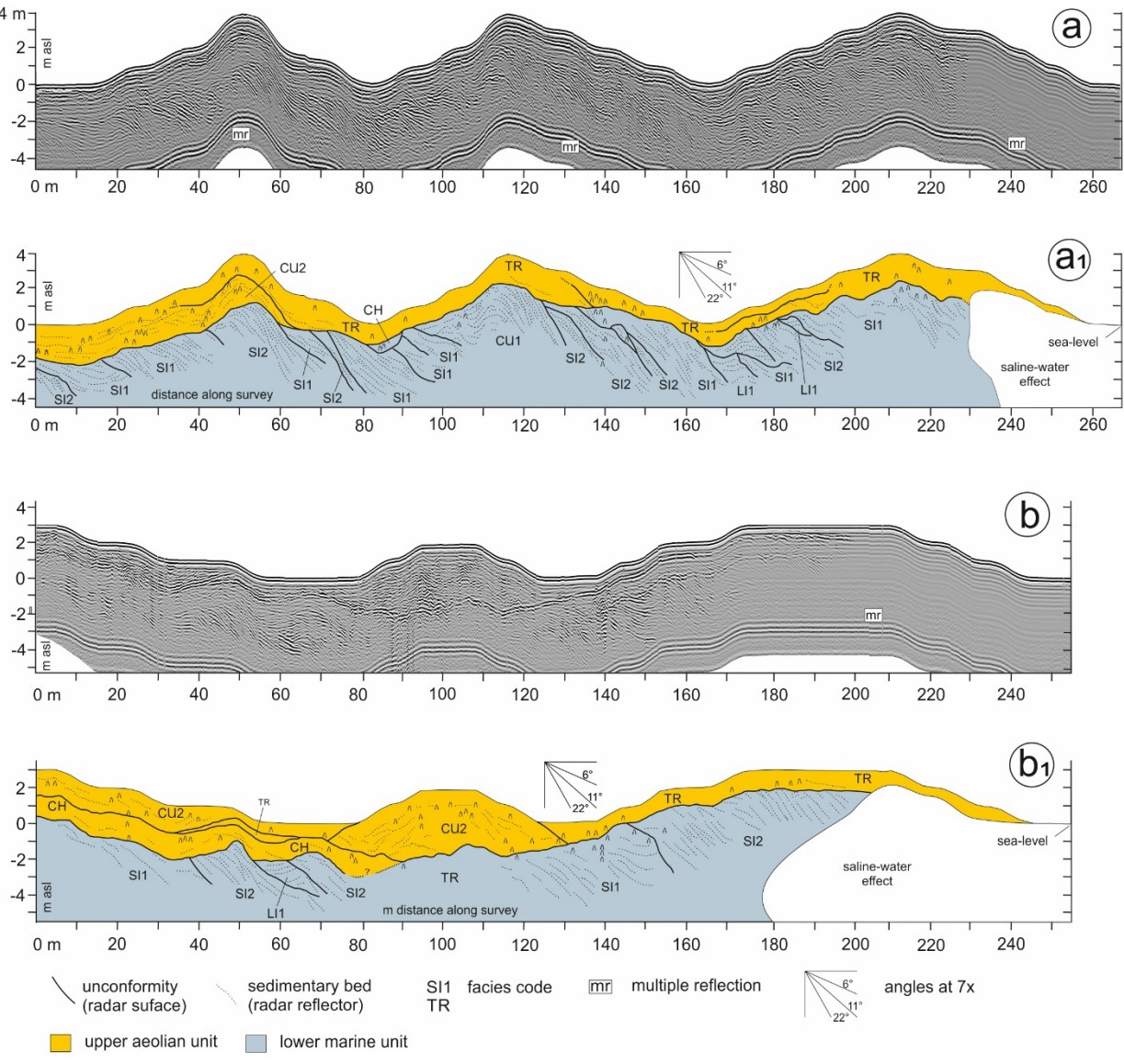

**Figure 6.** *Cont.*

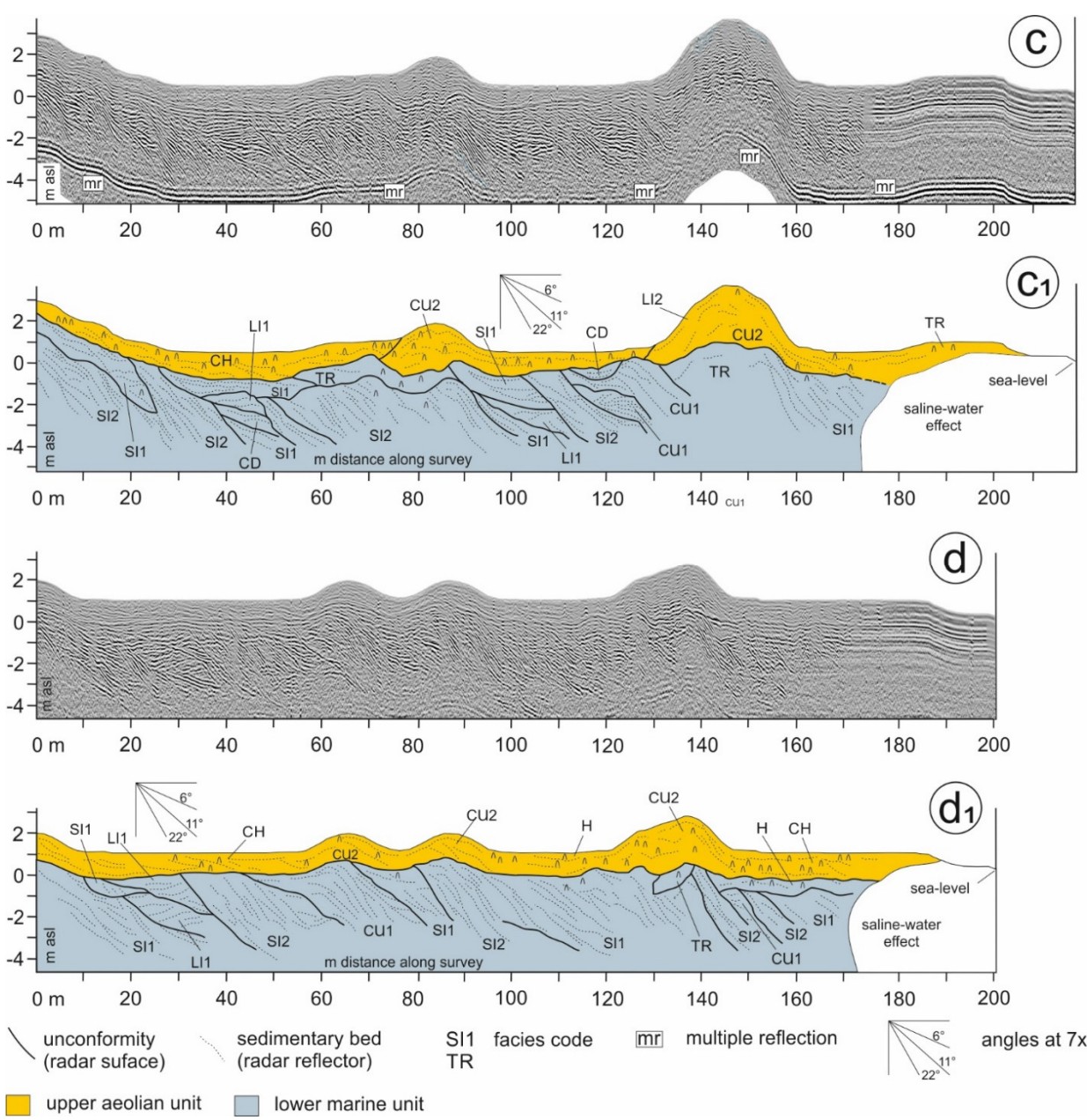

**Figure 6.** *Cont*.

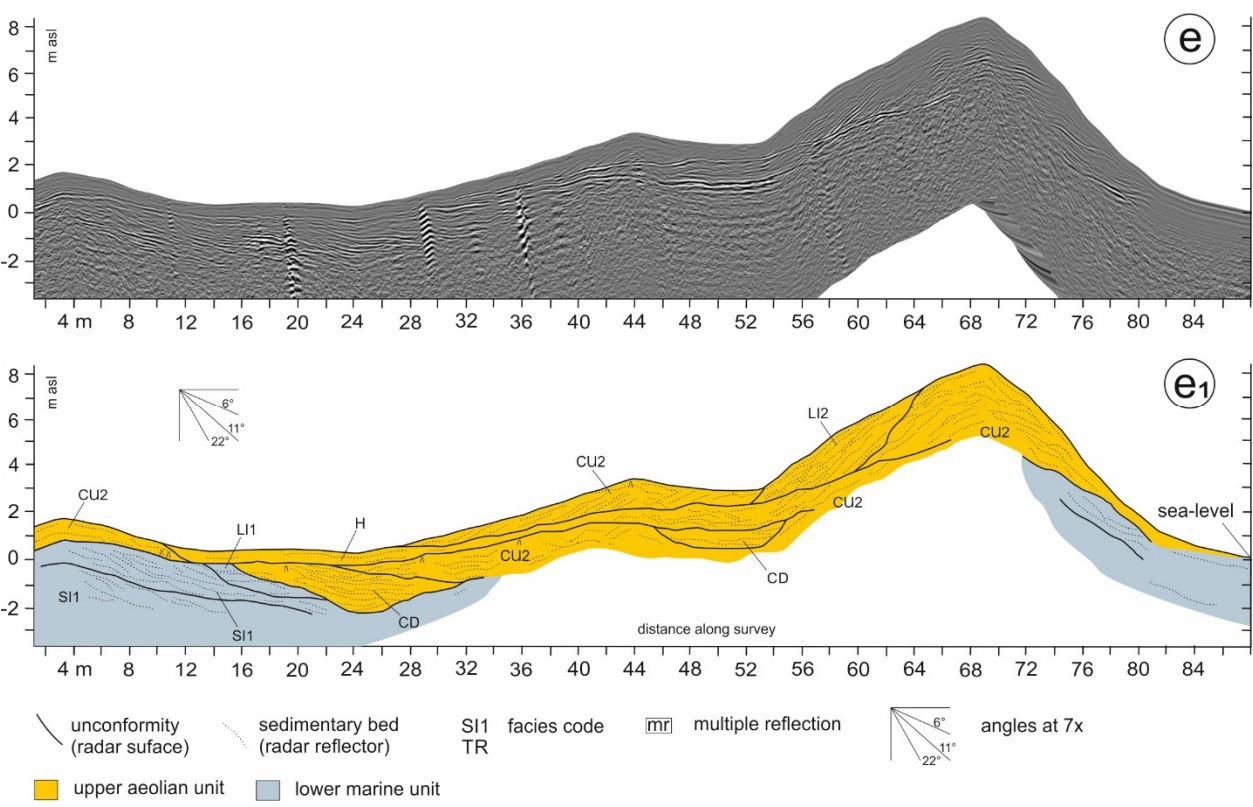

**Figure 6.** Processed (**a**,**b**,**c**,**d**,**e**) and corresponding interpreted (**a₁**,**b₁**,**c₁**,**d₁**,**e₁**) GPR profiles. For profiles location see Figure 2.

## 5. Discussion and Conclusions

Our radar survey showed that the analyzed sites experienced beach progradation that was not continuous but punctuated by the formation of berms. Other radar features point out a process of dune development starting from the formation of a berm during stormy weather that, due to return to fair weather along with the abundance of sediment supply, became inactive because it was no longer reached by the (storm) wave run-up. In the context of strand-plain progradation, the increase in distance between the beach face (foreshore) and the location of the storm berm prevents the aggradation of beds on the seaward side, and the overtopping of sedimentation onto the ridge and washover phenomena. Currently, the berm detached from the marine processes. The morphology of the relict storm berm, even if forming a low relief, interferes with the wind dynamics and can become the embryonic core of a wind-formed coastal dune. In this sense, the elevation of a coastal dune can correlate with the rate of coastline progradation.

A low rate of progradation allows the supply and activity of frontal dunes and their growth over time. This seems to be the case for the southern sector, where the elevation reached by the frontal dune, although affected by strong ongoing erosion processes, is up to 5–6 m (Figures 6 and 7a–d). This model, which has long been used, especially for sandy-gravel beaches ([3,11] and references therein), can also be used for sandy coasts such as the one examined, even if the inclinations and thicknesses of the beds have to be reconsidered according to the grain size and storm intensity.

We consider a positive sedimentary balance with a high rate of sedimentary supply to be the major reason for the increasing separation between the beach face and the berm. In the northern sectors, GPR results show that a wind-dominated phase followed the beach progradation phase. The strand-plain progradation was not a continuous process because erosive events (radar surface, i.e., erosional unconformities) interrupted the constructional processes. These erosive events can be related to temporary reductions of sedimentary nourishment or related to high-energy events such as storms. However, net strand-plain

progradation occurs because the rate of sedimentation on the beach's face prevails over periods of either no deposition or coastal erosion.

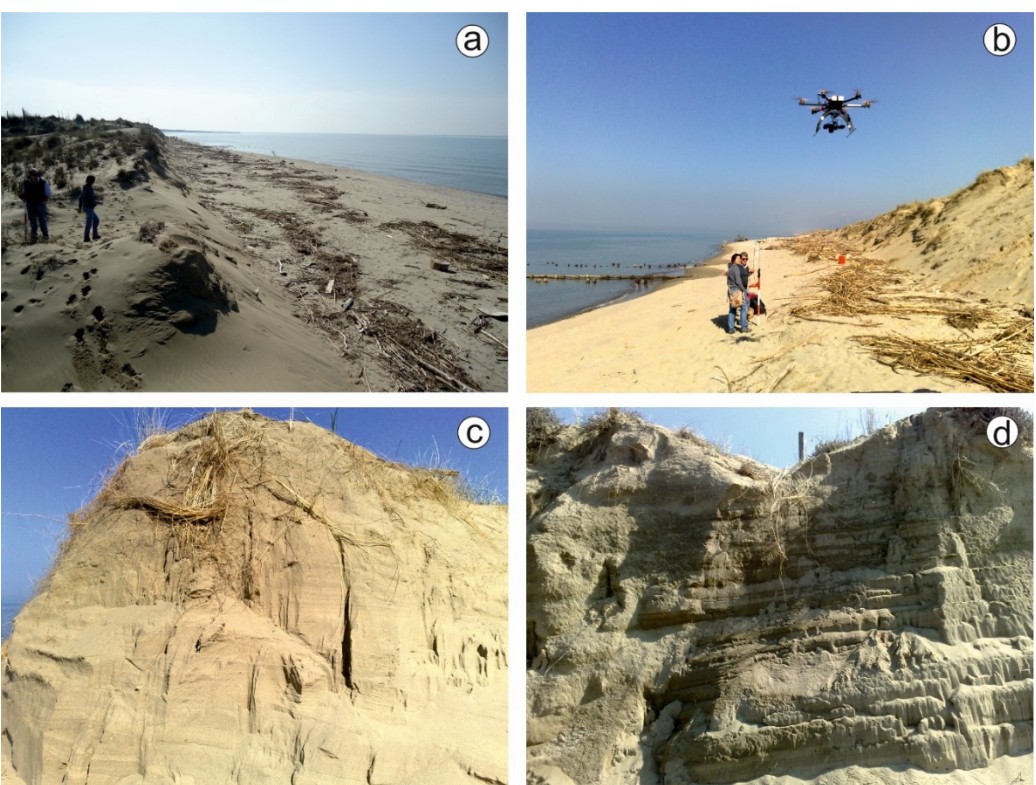

**Figure 7.** The geomorphology and sedimentary characteristics of the coastal stretch affected by erosion. The short backshore and the steep active dune flank (**a**,**b**), aeolian beds (**c**) and seaward inclined marine beds (**d**) exposed in natural trenches.

In the last 60 m of the profiles (from the beach face inland), the northmost sector (Figures 2a and 6) shows a large area constituted by an incipient dune with pioneer vegetation passing to a backshore. The extension of the area and the scarce development and growth of a frontal dune system are indicative of a high rate of progradation, as confirmed by several studies regarding the analysis of historical coastline locations ([33] and references therein). Conversely, a different morphodynamic framework characterizes the southern sector. There, strong erosional processes (Figure 7a,b), mainly due to the significant decrement in sediment supply [42], have taken place since the last century [34]. The distance between the foreshore and the base of the frontal dune is very short (~10 m, Figure 7a,b), and therefore, the seaward flank is affected by the wave run-up. The wave run-up has a destructive effect on the berm-dune system by creating erosional unconformities.

These data collectively show how the imaging of a strand-plain subsurface through a GPR survey can provide relevant elements supporting hypotheses on the temporal and spatial evolution of depositional settings. Therefore, the radar features may highlight the roles of the sedimentary budget in directly controlling the construction/destruction of backshore landforms and indirectly controlling the construction of a dune system [57,58]. The correspondence between the radar evidence (in this case related to recent times) and the ongoing morpho-sedimentary processes suggests that this approach can be used to reconstruct environmental history from further ago. We could explore greater depths and extend our investigations to more inland coastal areas. We believe the next investigations should include 3D-GPR prospections that can fully display the mutual relationships between facies and unconformities in the geometry in the three dimensions, strongly implementing the interpretation. However, stratigraphic investigations (i.e., boreholes and trenches) and chronological constraints (OSL and [14]C ages) are indispensable to verifying



the radar profiles' interpretations and providing the timing of the sedimentary processes involved.

**Author Contributions:** Conceptualization, A.R. and G.S.; methodology, A.R. and G.S.; validation, A.R., D.B., M.B. and G.S.; formal analysis, A.R. and G.S.; investigation, A.R., M.B.; resources, A.R.; data curation, A.R.; writing—original draft preparation, A.R.; writing—review and editing, A.R., D.B., M.B. and G.S.; visualization, A.R.; funding acquisition, A.R. All authors have read and agreed to the published version of the manuscript.

**Funding:** This research was funded by University of Pisa, PRA, 2020–21, "The termination I. The environmental and paleoclimatic variations occurred during the 25–11 ka period" (leader A. Ribolini).

**Institutional Review Board Statement:** Not applicable.

**Informed Consent Statement:** Not applicable.

**Data Availability Statement:** The data are available from the authors upon request.

**Acknowledgments:** The authors thank the Migliarino-San Rossore -Massaciuccoli Regional Park for the permission to enter protected areas; IDS GeoRadar (https://idsgeoradar.com/) for the instrumentation support; and Alberto Garzella and Lisa Zoia for fieldwork assistance.

**Conflicts of Interest:** The authors declare no conflict of interest.

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
