# Peer review of "Ground-Penetrating Radar Prospections to Image the Inner Structure of Coastal Dunes at Sites Characterized by Erosion and Accretion (Northern Tuscany, Italy)"

_applsci, doi:10.3390/app112311260_

Round 1

Reviewer 1 Report

You can find my comments in the attached file.
Kind regards

Author Response

Reviewer 1 –

REV- In my opinion the paper is written correctly and has a good quality to be published in the magazine.

AUTH - We thank the reviewer for appreciating the paper.

Some aspects that would have to be improved or modified:

REV: The Gramin barometric altimeter used has an accuracy of +/- 3m. I believe that an alternative method would have to be found (precision MDT, most accurate gps, etc ...) with better precision to verify the real error of this Garmin system and, if necessary, improve the topography of the GPR profiles before applied topographical correction.

AUTH – We are aware that +/- 3m is relatively poor accuracy. To partially overcome this problem, we use the GPS in "barometric mode" and not in "satellite mode" when acquiring the elevations along the survey profile. By rapidly acquiring a topographic profile during the survey (i.e. with a stable barometric pressure value) we are quite sure that the altitudes are affected by a fairly constant inaccuracy between measurements. This means that the entire profile is most likely not accurate but consistent (all measurements should be shifted up or down by the same amount). The last measurement along every profile is made at sea level. A value of 0 m asl was then imposed on this measurement, and the difference between the recorded measurement and 0 m asl was used to correct all other measurements of the profile.

However, given the purposes of the work, we do not need to relate the elevation of some GPR reflectors or radar facies with respect to sea level for paleoenvironmental reasons (e.g. to speculate about former sea levels). Therefore, we think that the use of a topographical profile representing the surface in a reasonable way can be accepted.

REV - It is write in the text that the processing speed is calculated from the hyperbolas that appear in the profiles, this text appears in the section 3. Materials and Methods. It would have to show these structures used and the calculation of the propagation speed, maybe in a new figure.

AUTH – This is a good point because this procedure is quite common in GPR documents, but we understand that Applied Sciences readers may not be confident with the method. We have added a figure to illustrate this.

REV- I suggest redraw figure 3 in table form. It should be possible to consult the different types of radarfacies more easily, in a form like that….

AUTH – We thank the reviewer; Figure 3 has been updated accordingly. We prefer to insert only horizontal lines just so as not to complicate the figure with too many lines. To make the set of facies more evident, we have added an alternation of gray and white backgrounds.

REV - The radarfacies labels in figures 5-6-7 are extremely small, they would have to be enlarged.

AUTH – Thanks for the note. We have increased the font of the facies codes.

Reviewer 2 Report

Dear Authors,

I'll start with the title: Ground-Penetrating Radar prospections to image the inner structure of coastal dunes at sites characterized by erosion and accretion = increase (northern Tuscany, Italy) - the word accretion is not appropriate in this case, it refers to geology. In the case of such research, we use the word increase. I noticed other than accretion in the text.

Your work is interesting but needs improvement. I have posted my comments below, on each point and in the submitted article.

  1. The introduction should be supplemented by at least Ludwig et al. 2017. The literature review is weak, you focus your attention on older works, and newer works have been created. It is generally known that GPR was used to analyze sandy sediments, but also to identify other sediments (Kramer et al. 2012, Åšnieszko et al. 2017, etc.).
  2. Study Area
  3. Methods - good point, well described, only as the conclusions show, there are no wells,
  4. Results - in the references in the text to Figures 5, 6 and 7, I propose to add symbols to which segments they refer. The reference to several figures at the same time is not correct without indicating a specific, marked figure.
  5. Discussion and conclusions - please separate both points. Besides, please take a stance on the analyzes of other researchers, and do not use the abbreviations references therein. 
  6. Conclusions - please expand them.

Best regards

Reviewer 

Author Response

Reviewer 2 –

REV - I'll start with the title: Ground-Penetrating Radar prospections to image the inner structure of coastal dunes at sites characterized by erosion and accretion = increase (northern Tuscany, Italy) - the word accretion is not appropriate in this case, it refers to geology. In the case of such research, we use the word increase. I noticed other than accretion in the text.

AUTH – Thanks for the comment. We agree, “accretion” is a word normally used in describing a coastal progradation from a geological point of view, specifically dealing with stratigraphical and geomorphological features. However, our paper is based on the application of the geophysical method in a geological context, which is an example of applied science. Moreover, the title of the Special Issue (and the motivations) clearly refers to GPR application to near-surface geology, geomorphology etc.. We understand that “accretion” may sound like a weird word, but we prefer to maintain it.

REV - Your work is interesting but needs improvement. I have posted my comments below, on each point and in the submitted article.

AUTH – We thank the reviewer and report below our replies to his/her comments.

REV - The introduction should be supplemented by at least Ludwig et al. 2017.

AUTH – Thanks for the suggestions, but we have reported the reference of Lindhorst and Betzler 2016 that is essentially very similar to Ludwig et al 2017 as regards research design, method, data processing, and interpretation.

REV - The literature review is weak, you focus your attention on older works, and newer works have been created. It is generally known that GPR was used to analyze sandy sediments, but also to identify other sediments (Kramer et al. 2012, Åšnieszko et al. 2017, etc.).

AUTH – The purpose of the literature review was to bring to the attention of readers important papers on the study of coastal sediments in contexts similar (or not far away) to the one we investigated. In our opinion, it is not necessary to inform the reader about the application of GPR to different contexts. In addition to the fact that the most recent papers are not always the best, the GPR references in the Introduction concern known papers (in some cases outstanding). We prefer not to implement the reference list.

REV - Methods - good point, well described, only as the conclusions show, there are no wells,

AUTH – Thanks, yes the calibration that would be allowed by the boreholes is missing, in fact this is the next point on our agenda.

REV - – Results - in the references in the text to Figures 5, 6 and 7, I propose to add symbols to which segments they refer. The reference to several figures at the same time is not correct without indicating a specific, marked figure.

AUTH - We understand this point, but the issue is that what is reported in the text is evident in all the cited figures. However, to be clearer we have added the letter of the panel of the figure to differentiate at least when the reference is to uninterpreted or interpreted profiles.

REV - Discussion and conclusions - please separate both points.

AUTH - Applied Sciences offers the option not to include a conclusion paragraph. We preferred to bring Discussion and Conclusion together because in the flow of the discussion of the results there are also some considerations that may sound like conclusive. Separating the two paragraphs means writing an extremely short Conclusion or simply listing some concepts expressed earlier in the text. Both solutions are, in our opinion, useless for the reader.

REV - Besides, please take a stance on the analyzes of other researchers, and do not use the abbreviations references therein.

AUTH - The expression ".... and reference therein" is classically used in literature not because other authors cannot be cited, but to say that in the last citation the reader can find many older references, so as to avoid a long list of names (numbers in this case).

REV - Conclusions - please expand them.

AUTH - Honestly, we believe that the concepts reported in the Discussion / Conclusion paragraph are representative of the results of the work and their relevance in the context of the evolution of a dune-beach system. We are not inclined to write a Conclusion just to have a new paragraph in the list.

AUTH -We thank the reviewer for the edits to the PDF, we have changed the text accordingly.